# Study on the performance of Type II vertical rescue timber shores based on experiment and simulation

**Feng Zhang**[ID]*, **Xiangyang Lu, Hao Zhang**

Shandong Earthquake Emergency Service Center, Shandong Earthquake Agency, Jinan Shandong, China

* 2019071041@cauc.edu.cn

## Abstract

Structural instability poses significant risks at earthquake rescue sites, where Type II vertical timber shores play a pivotal bridging role in rapidly securing operational space safety. To systematically evaluate their structural performance and failure mechanisms, this study employed a self-developed hydraulic testing apparatus to conduct compressive bearing capacity tests on three Type II vertical shore configurations: Double T-A, Double T-B, and Two-Post. Vertical downward displacement-controlled loading was applied until failure. Concurrently, refined finite element models incorporating key joint contact characteristics (wedges, plywood gussets, nails) were established using Abaqus software for numerical simulation. Experimental and simulation results demonstrated strong agreement, revealing that: The Two-Post shore exhibited the highest ultimate bearing capacity (239.0 kN), significantly outperforming the Double T-B (200.8 kN) and Double T-A (190.9 kN) types. Failure was typically preceded by audible cracks from wedge cracking, plywood gusset expansion, and buckling of posts near the mid-span. The yield point was identified on the load-displacement curves via the farthest point method. Integrating the Allowable Stress Design (ASD) method and site-specific human-machine-environment factors (psychological impact coefficient $C_p$, wood moisture defect coefficient $C_m$, load duration coefficient $C_d$), a safety factor $K = 2.0$ is proposed for calculating the design bearing capacity ($F_d = F_u / 2.0$). This research elucidates the failure mechanism of Type II vertical timber shores, confirms the superiority of the Two-Post configuration, emphasizes the critical influence of load centering on support effectiveness, and provides essential experimental data and a theoretical basis for the safe design and application of timber shoring systems in earthquake rescue operations.

**Data availability statement:** Data cannot be shared publicly because of confidential. Data are available from the Shandong Earthquake Agency Data Access (contact via 885071337@qq.com) for researchers who meet the criteria for access to confidential data.

**Funding:** Supported by the General Research Project (YB2410) of Shandong Earthquake Agency.

**Competing interests:** The authors have declared that no competing interests exist.

## 1. Introduction

In Urban Search and Rescue (US&R) operations, preventing further structural collapse is paramount for site safety [1–3]. Emergency Shoring Systems (ESS), particularly Wood Shoring Systems (WSS), are crucial technical measures for stabilizing damaged structures, mitigating risks to rescuers and victims [4–6]. WSS, favored for material availability and low cost, was first formally applied by FEMA US&R teams in 1989 and subsequently used in events like the 1995 Oklahoma City bombing and the 2001 World Trade Center collapse [7,8]. Recently, WSS was effectively deployed to stabilize structures during the 2022 Changsha building collapse rescue [9].

Standardized timber shores function by transferring loads from damaged structures to other load-bearing elements, redistributing forces, and resisting collapse from aftershocks. Thus, they must meet requirements for dimensional adjustability, tight component connections, lateral stability, material ductility, and overload warning. As shown in Fig 1, vertical timber shores are classified into three types: Type I (one-dimensional), Type II (two-dimensional), and Type III (three-dimensional) [10]. Type I shores are used for initial temporary propping. Type II shores are constructed nearby to rapidly secure safe operational spaces within collapsed structures. When severely damaged areas necessitate Type III shores, Type II shores provide initial temporary support before being grouped and converted. Consequently, Type II shores serve a critical bridging function within the WSS framework.

Existing research on Wood Shoring Systems (WSS) has established a foundational yet incomplete understanding of their structural behavior. Early investigations, such as the practical work by O'Connell [4], laid the groundwork for field applications. To address the lack of standardized protocols for timber shoring in rescue operations—despite its recognized importance—Tang developed a structured, team-based methodology. His research, centered on the Bill of Materials, operational procedures, and application essentials, provides detailed guidelines for four key scenarios [10]. Subsequent evaluations by early US&R teams revealed performance differences among configurations, noting that plywood-gusseted Type III shores exhibited a 17%−37% lower capacity than raker-braced types, prompting initial design optimizations [7,8]. With the advancement of numerical tools, scholars like Liu [11] and Xing [12] conducted static finite element analysis (FEA) on typical shores using ANSYS, successfully identifying global deformation zones. Using Midas Civil software, Zhang et al. simulated timber shores under static loading to investigate their structural behavior [13]. The analysis of axial, shear, and bending moment results revealed their mechanical characteristics and vulnerable zones. However, their models oversimplified or entirely neglected the critical influence of key joints (e.g., wedges, gussets, nails). A step forward was taken by Xu [14], who incorporated joint behavior via spring models, achieving partial agreement with experimental results. Nevertheless, this approach still lacked detailed stress-strain visualizations and a comprehensive representation of joint mechanics. Collectively, these prior studies are characterized by three primary limitations that define the research gaps addressed in this work:

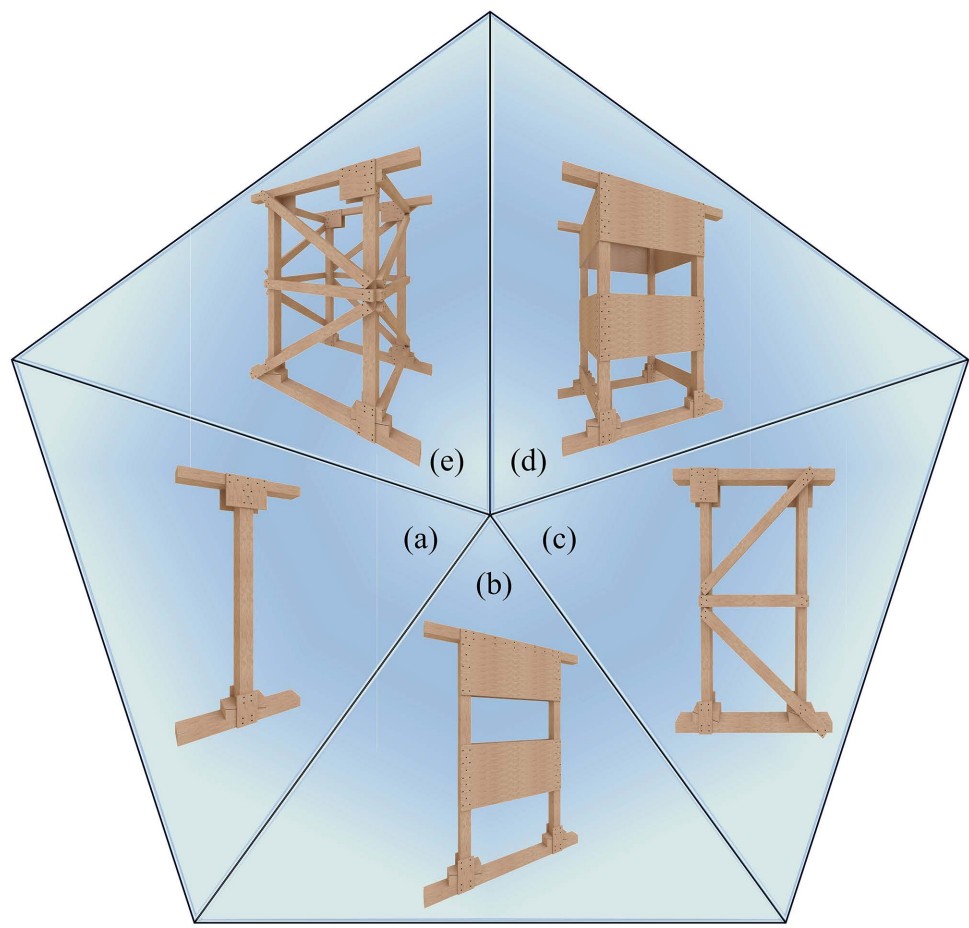

**Fig 1. Classification of vertical timber shoring systems:(a) Type I single-T shore, (b) Type II double-T shore, (c) Type II two-post shore, (d) Type III plywood-gusseted 3D shore, (e) Type III raker-braced 3D shore.**

a) A predominant reliance on load-controlled testing methods, where continuous loading to failure often leads to rapid, catastrophic collapse, thereby hindering the observation of the complete evolution of failure characteristics.

b) Widespread simplification or neglect of key joint details in numerical simulations, which compromises the accuracy of stress distribution predictions and failure mechanism analysis.

c) The absence of an integrated design methodology that synergistically combines experimental data, high-fidelity simulation, and operational safety factors tailored for the unique demands of rescue shoring.

To address the documented limitations, this study introduces an integrated approach whose novelty is manifested in three aspects: first, the adoption of displacement-controlled testing to fully capture failure progression; second, the development of refined FE models that explicitly represent key joints; and third, the introduction of a comprehensive safety factor accommodating structural and human-environmental factors. Experimentally, a self-developed hydraulic setup applies a displacement-controlled load to three shore types, with synchronous recording of visual/auditory cues serving as early warnings. Numerically, refined Abaqus models are developed and validated against experimental capacity curves and stress contours, thereby providing valuable data and practical guidance for implementing these shores in earthquake rescue.

## 2. Materials and methods

### 2.1. Type II shore specimens

Based on NFPA 1670:2022 (Standard on Operations and Training for Technical Search and Rescue Incidents) standards, two Type II shore models were selected: Double T Shore (DT) and Two-Post Vertical Shore. The Double-T (DT) shore is a foundational Type II design, and we subdivided it into DT-A (without a large central gusset) and DT-B (with a large central gusset) to explicitly investigate the effect of this key structural variation on performance. The Two-Post configuration represents another common, distinct structural system within the Type II classification, allowing for a direct comparison between "frame" (Two-Post) and "braced panel" (DT) typologies. This selection covers the primary structural variants of Type II shores used in rescue operations. As depicted in Fig 2, all three types shared unified dimensions (overall height 180 cm, width 120 cm), with posts symmetrically placed along the centerline (60 cm inner edge spacing). Component sections were standardized: posts and sole plates (10 cm × 10 cm), horizontal and diagonal braces (10 cm × 5 cm), wedge blocks (10 cm × 10 cm × 30 cm), plywood gusset thickness (1.3 cm; half gusset 15 cm × 30 cm; full gusset 30 cm × 30 cm; large gusset 30 cm × 80 cm). All head plates, posts, and sole plates were secured with plywood gussets and braces connected via nails (penetration depth ≥ 1/2 nail length). Specimens used Scots pine (Pinus sylvestris var. mongolica) with a density of 0.5 g/cm³ and 15% moisture content. Installation involved straightening and pre-tensioning the system using wedges, followed by fixing half gussets before testing. The tests were conducted on three identical specimens for each of the three shore types (DT-A, DT-B, Two-Post), resulting in a total of nine tests. The reported values (e.g., ultimate load) are representative of the consistent results observed across these replicates.

### 2.2. Testing platform

The testing apparatus (Fig 3) comprised a rigid frame adaptable to different shore heights. The procedure involved three stages: (1) Positioning the loading plate precisely centered on the shore specimen using a chain hoist. (2) Transferring the hydraulic load through the loading plate to the specimen and then to the reaction base. (3) Applying a displacement-controlled vertical load at a constant rate (minimum 0.05 mm/s) via a hydraulic servo system (50 t max force, 500 mm stroke) until peak load and failure. The loading rate of 0.05 mm/s was selected as a quasi-static rate to

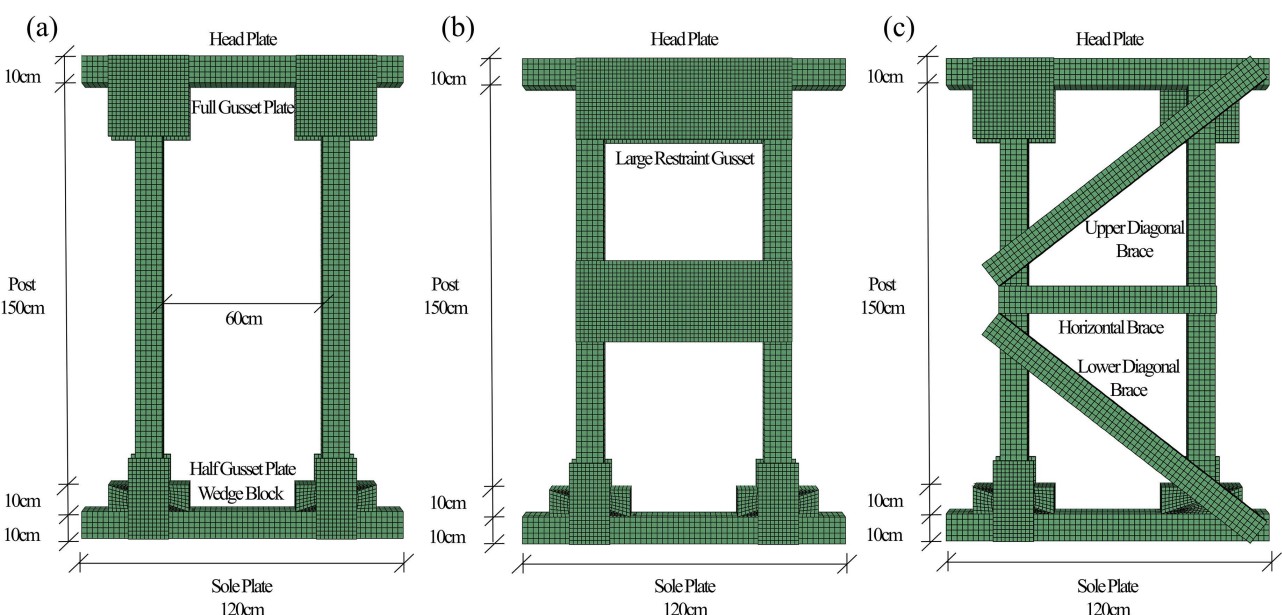

**Fig 2. Classification of Type II vertical rescue timber shores:(a) DT-A, (b) DT-B, (c) Two-Post.**

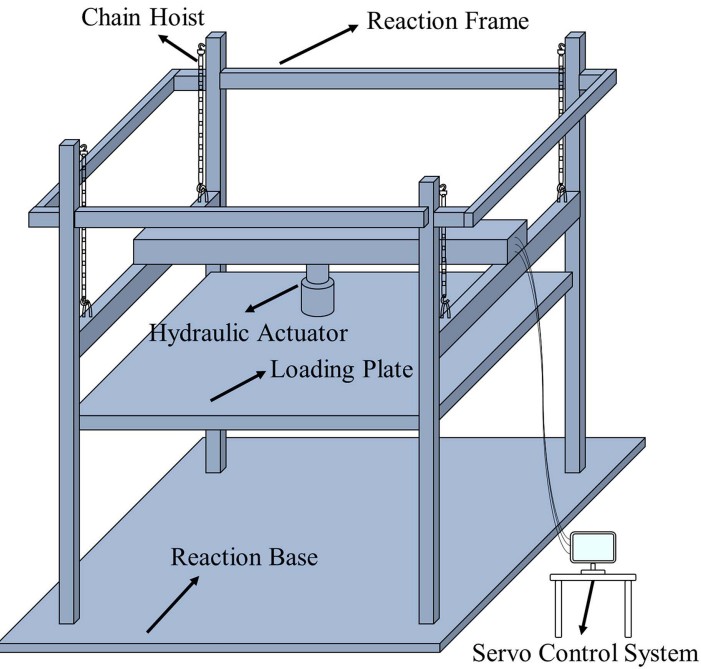

**Fig 3. Test setup for compressive bearing capacity of vertical shores.**

ensure stable and controllable propagation of damage, allowing for clear observation of failure sequences (e.g., wedge cracking, gusset expansion) which would be challenging to capture under faster, load-controlled scenarios typical in prior studies. This rate is consistent with standards for quasi-static testing of wood structures (e.g., ASTM D143, Standard Test Methods for Small Clear Specimens of Timber).

## 2.3. Simulation conditions

Finite element models were developed in Abaqus, referencing validated methods for wood compression and joint deformation [15–17]. The isotropic assumption was a simplification made primarily for computational efficiency in this initial, comparative nonlinear analysis focusing on global structural response and failure modes rather than highly localized wood grain effects; plywood gusset-nail and brace-nail assemblies were modeled as monolithic parts. Structured meshing was employed, with critical regions (head plates, posts, wedges) refined to 15 mm and others set to 20 mm. General contact was defined on all external surfaces: "Hard Contact" in the normal direction (separation allowed), and a "Penalty" friction model (coefficient 0.15) in the tangential direction. Nail connections were simulated using "Tie" constraints. The sole plate was fully fixed ($U_1 = U_2 = U_3 = UR_1 = UR_2 = UR_3 = 0$), while the head plate was constrained horizontally and in rotation, with a vertical displacement applied in increments (0.001 m) via a reference point coupled to its top surface. Load-displacement data was extracted from this reference point.

## 3. Results and discussion

### 3.1. Numerical response analysis of type II vertical rescue timber shores

Under monotonic vertical loading, equivalent stress contours (Figs 4–6) revealed that high-stress zones in all three shore types predominantly concentrated on the posts at ultimate load, with significant stress extending to wedge blocks, plywood gussets (half/full/large), and brace joint regions.

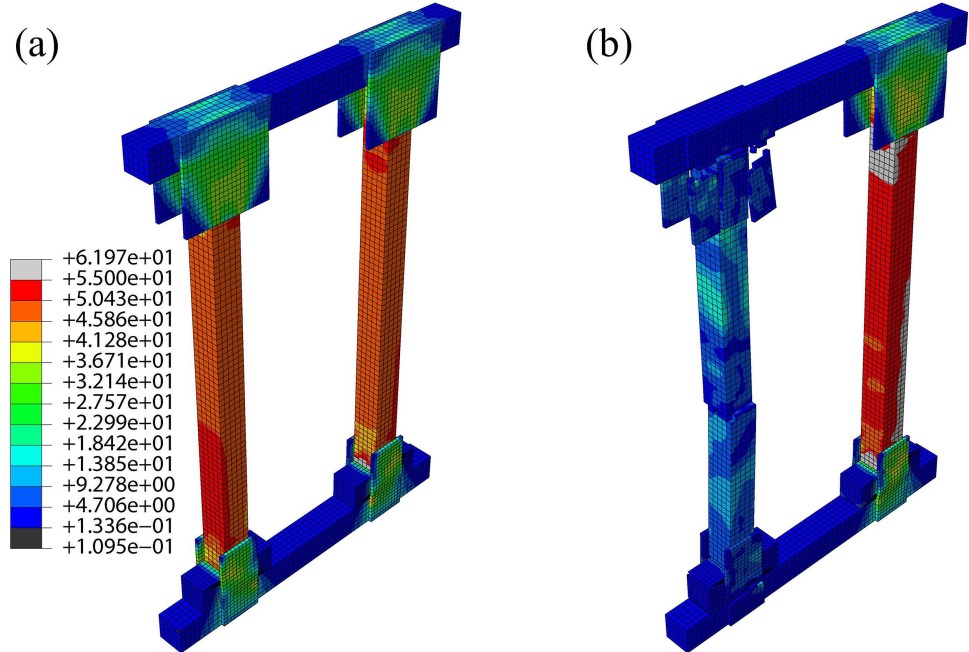

**Fig 4. DT-A shore:(a) Equivalent stress contour at ultimate state, (b) Equivalent stress contour at failure state.**

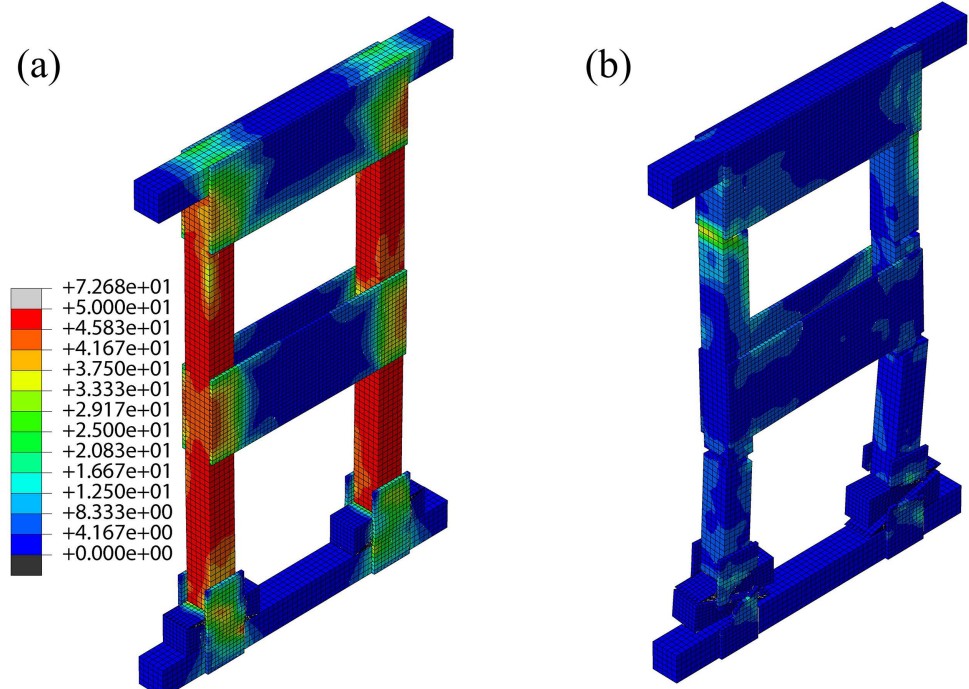

**Fig 5. DT-B shore:(a) Equivalent stress contour at ultimate state, (b) Equivalent stress contour at failure state.**

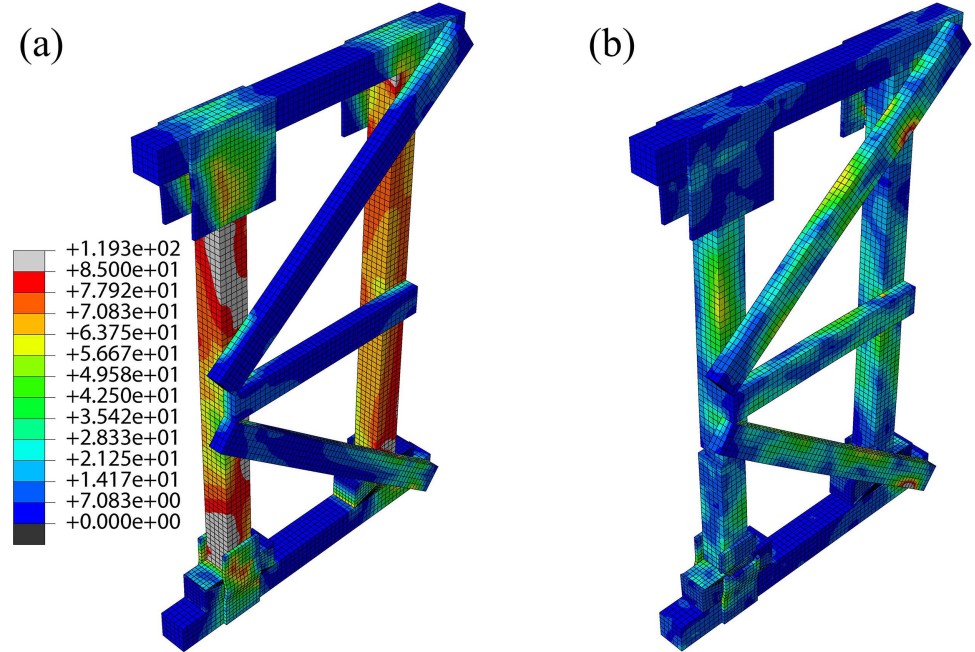

**Fig 6. Two-Post shore:(a) Equivalent stress contour at ultimate state, (b) Equivalent stress contour at failure state.**

Distinct stress concentration patterns were observed among the systems. In DT-A and DT-B types, load was initially shared relatively uniformly between the two posts, but severe stress concentration subsequently developed at the wedge-post interfaces and bottom half-gusset connections. The large central gusset in DT-B attracted and partially redistributed this stress. In contrast, stress in the Two-Post system was more localized on the individual posts of the braced frame, with critical concentrations at the mid-span third (the initiation point of buckling) and the connections with the "K"-braces. Although the head and sole plate gussets also experienced high stress, their influence was less dominant than in the DT types.

The bearing capacity ranked as: Two-Post > DT-B > DT-A, with marked differences in failure modes. DT-A/B types exhibited bottom half-gusset expansion and a symmetrical post stress distribution at ultimate load. Under increasing load, DT-B failed through simultaneous bilateral post buckling fracture subsequent to wedge separation and gusset fracture—a process constrained by its large central gusset. DT-A, however, failed via single-post fracture. The Two-Post shore showed more pronounced stress concentration on its left post. Following gusset expansion and fracture, the left post buckled and failed at the mid-span third; nevertheless, this configuration retained superior structural integrity post-failure compared to the DT types.

This performance hierarchy is primarily attributed to the "K"-braced frame system of the Two-Post shore, which establishes a more direct and efficient load path while providing enhanced redundancy and lateral restraint. The central large gusset in DT-B improves stiffness but does not fundamentally alter the basic DT load path mechanism. Simulations consistently identified posts, wedges, and plywood gussets as the primary failure risk zones, with post fracture being the direct failure cause, thus providing a theoretical basis for experimental failure analysis.

### 3.2. Experimental failure mechanism and simulation validation

The failure processes observed in compressive tests (Figs 7–9) showed close agreement with simulation results. Good consistency was achieved in global structural response (load–displacement curves) and failure modes, though local stress

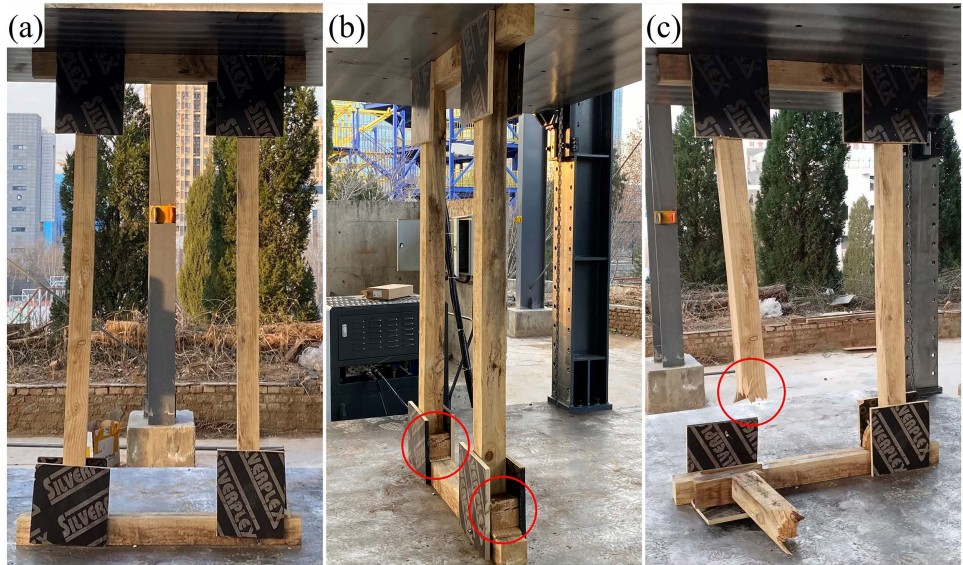

**Fig 7. Experimental failure process and local damage characteristics of DT-A shore.**

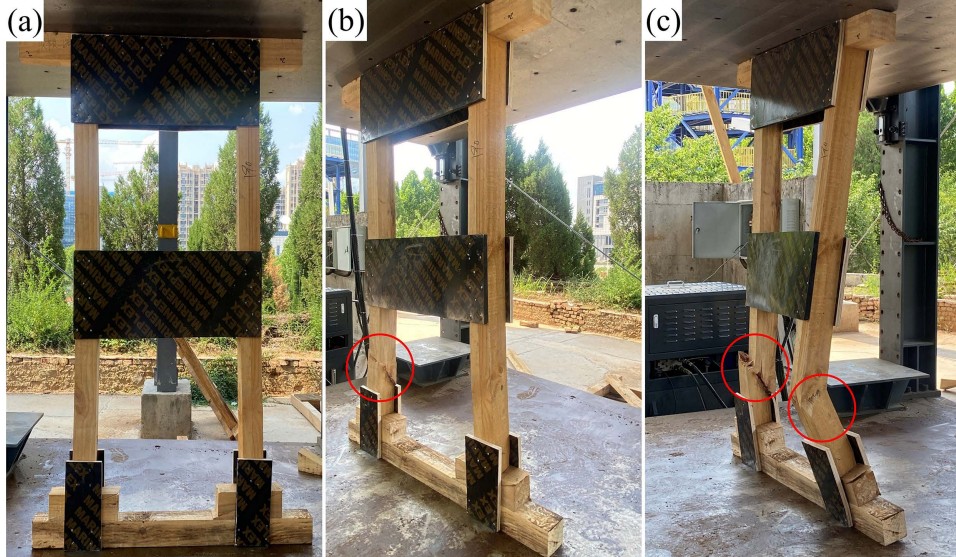

**Fig 8. Experimental failure process and local damage characteristics of DT-B shore.**

magnitudes were more sensitive to material heterogeneity and modeling simplifications such as the isotropic assumption. Accordingly, while the high-stress zones (posts, wedges, gussets) and their activation sequence were accurately reproduced, the absolute simulated stress values must be interpreted with caution in light of these modeling limitations.

During the initial loading phase, audible failure warnings were emitted from contact points, followed by transverse cracking and warping of the wedges due to excessive compressive stress. Subsequent behavior diverged among the types: DT-A exhibited bending and complete fracture of a single post at the lower third of the mid-span. DT-B underwent

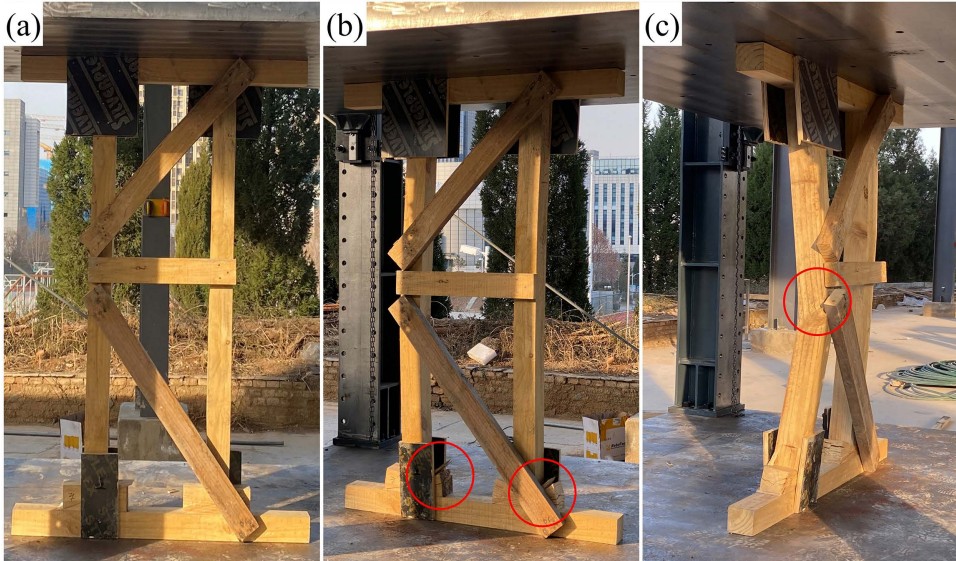

**Fig 9. Experimental failure process and local damage characteristics of Two-Post shore.**

single-post fracture→temporary realignment of the fractured section constrained by the large gusset→sequential fracture of both posts. The Two-Post shore experienced plywood gusset expansion, followed by buckling-induced fracture of a post at the mid-span third, while retaining superior residual structural integrity.

Quantitative analysis (Fig 10) indicated wedge compression heights in the order: Two-Post (7.0–7.3 cm) <DT-B (8.0–8.5 cm) <DT-A (8.3–8.8 cm), confirming the most pronounced plastic deformation in the Two-Post wedge. The failure mechanisms can be summarized as follows: DT types underwent a chain reaction of "contact surface compression →

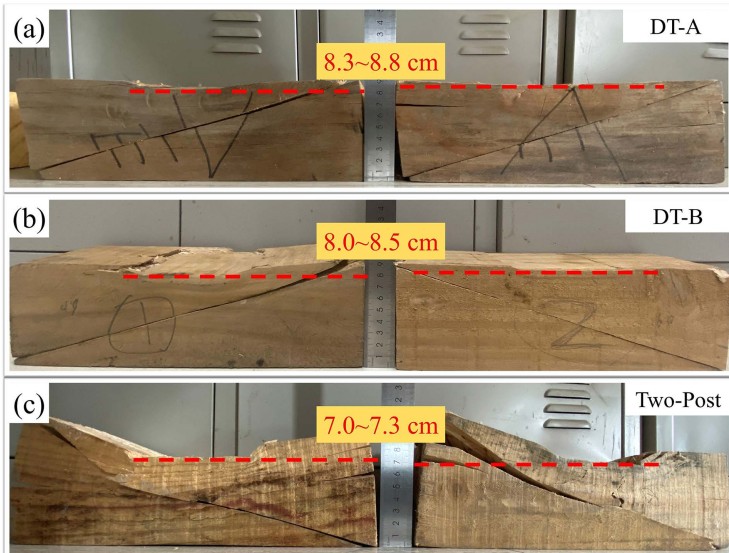

**Fig 10. Wedge failure characteristics:(a) DT-A, (b) DT-B, (c) Two-Post.**

wedge cracking/warping → post bending → fracture at the lower third of the mid-span," whereas the Two-Post type incorporated an additional plywood gusset expansion phase, shifting the failure location to the mid-span third. The fracture of plywood gussets—particularly the bottom half-gussets in DT types and the head/sole plate gussets in the Two-Post type—served as a key trigger for system instability. This fracture represents the loss of lateral restraint at critical joints, leading to increased eccentricity and lateral displacement of the posts, which rapidly progresses to post buckling and catastrophic collapse. It is thus not merely a localized weakening, but a pivotal event in the overall failure sequence.

The performance superiority of the Two-Post shore originates from its "K"-braced frame system, which effectively reduces the brace angle and distributes horizontal forces, thereby enhancing both load-bearing capacity and post-failure integrity. Although the residual capacity after main post fracture was not quantitatively measured, visual observation (Fig 9c) and simulated stress contours (Fig 6b) clearly demonstrated that the "K"-braced frame of the Two-Post shore maintained its geometric configuration substantially better than the fully collapsed DT types. This suggests a potentially higher residual load-bearing capacity and, more critically, a less abrupt and more ductile failure mode, affording a vital additional safety margin for personnel beneath the structure. The tests successfully replicated the high-risk zones predicted by simulations, validating the model's accuracy in capturing wood brittle fracture behavior.

### 3.3. Bearing capacity analysis of Type II vertical rescue timber shores

Load-displacement curves (Fig 11a-c) revealed significant differences: DT-A and DT-B had ultimate capacities ($F_u$) of 190.9 kN and 200.8 kN, respectively, while the Two-Post type reached 239.0 kN. Curve evolution exhibited three distinct phases: (1) An initial system engagement zone with increasing stiffness, attributed to the gradual closure of prefabricated component gaps (post-wedge-head/sole plate) constrained by nailed gussets. (2) A full engagement zone with constant stiffness, reflecting all components working synergistically under compression, maximizing system stiffness. (3) A failure zone characterized by stiffness degradation and a sharp post-peak load drop, induced by wood plastic deformation and loss of single-shear joint stiffness leading to structural instability.

To determine the structural yield point ($F_y$) for safe design, the farthest point method was applied to discrete data (Fig 11d) [18,19]: A line Ou is drawn from the origin O to the ultimate point $u$; the point $y$ where a line parallel to Ou is tangent to the curve defines the yield point. The resulting compressive mechanical model (Fig 12) indicates the ascending curve can be simplified as a bilinear model: an elastic phase ($O{\rightarrow}y$) with linear response, and a plastic phase ($y{\rightarrow}u$) accompanied by progressive wedge warping and fracture, culminating in post buckling fracture under peak load causing overall failure.

Based on the Allowable Stress Design (ASD) method [20], the relationship between the design bearing capacity ($F_d$) and the ultimate bearing capacity ($F_u$) for Type II vertical rescue timber shores is established as follows:

$$F_d \ = \ F_u \ / \ K$$

$$K \ = \ C_p \ \times \ C_m \ \times \ C_d$$

where: $K$ is the safety factor, integrating human reliability, material state, and environmental conditions at rescue sites. $C_p$, the psychological impact coefficient, is a conceptual factor derived from experimental observations where audible cracking sounds from wedge cracking induced rescuer stress. This factor links a structural warning signal (analogous to a yield point) to its potential operational implications, but it is not a rigorously validated psychological metric. $C_p$ is defined as the ratio $F_u/F_y$ (yield load), calculated as $C_p = 1.25 \sim 1.56$ across shore types. $C_m$ is the wood moisture defect coefficient, introduced to account for potential strength loss due to high moisture levels in real-world rescue environments (e.g., rain, humidity). This coefficient is derived from the US National Design Specification (NDS) for Wood Construction (2018 Edition) [21], which stipulates a reduction in bending strength to 0.85 times the dry value when moisture content exceeds

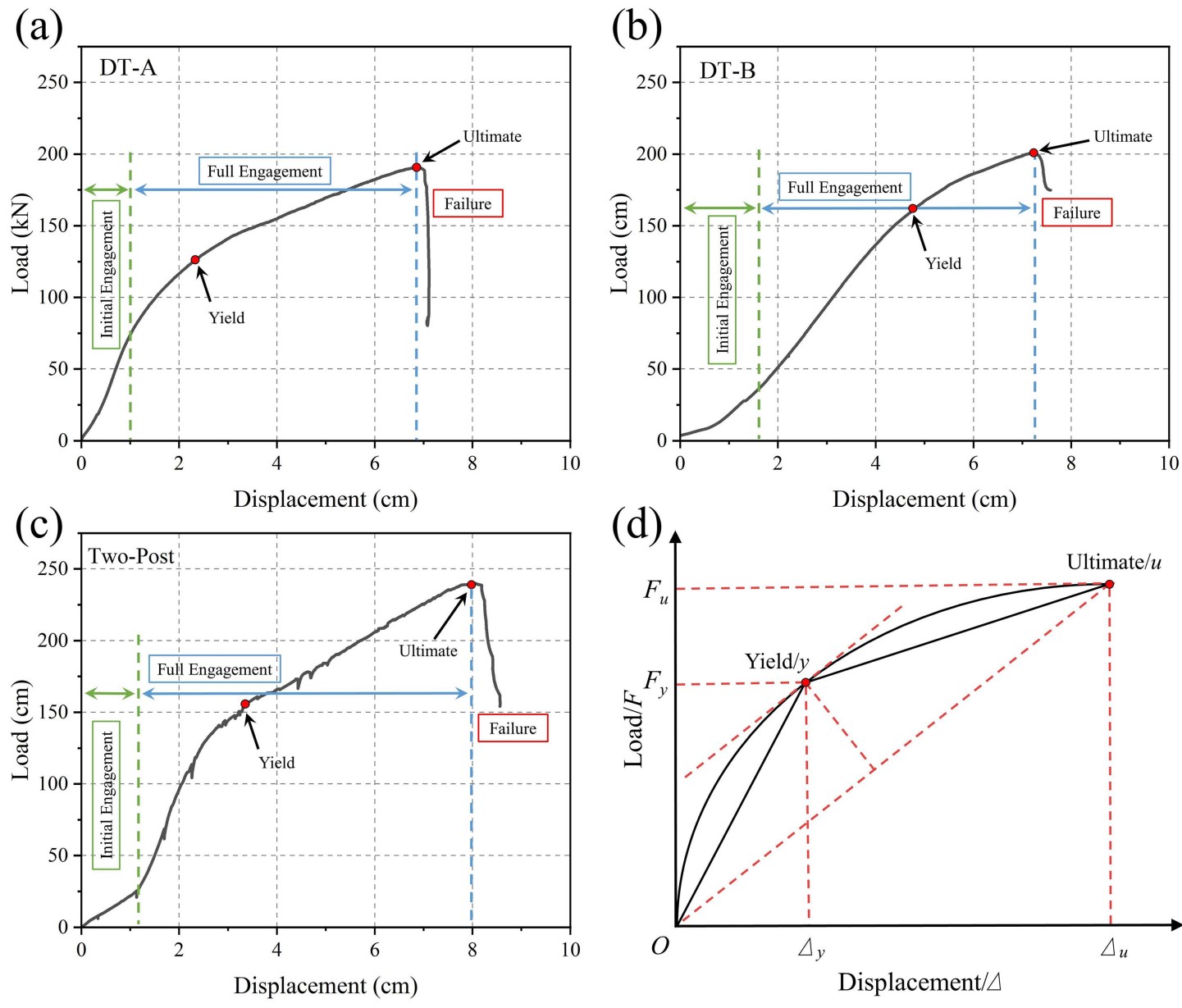

**Fig 11. Compressive load-displacement curves:(a) DT-A, (b) DT-B, (c) Two-Post;(d) Schematic of the farthest point method.**

19%. The inverse of this factor ($1/0.85 \approx 1.176$) defines $C_m$. To ensure a conservative safety margin, this value is rounded down to 1.15. $C_d$ is the load duration coefficient: For heavy rescue loads lasting 7–10 days, $C_d = 1.25$ is adopted based on the NDS long-term load provision. Synthesizing the safety factor: $K = (1.25 \sim 1.56) \times 1.15 \times 1.25 = 1.80 \sim 2.24$. Balancing safety margin and engineering feasibility, $K = 2.0$ is recommended. Thus, the design bearing capacity is: $F_d = F_u / 2.0$.

## 4. Working principle and application recommendations for Type II Vertical rescue timber shores

Type II vertical timber shores serve a critical function in collapsed building search and rescue operations by enabling rapid stabilization of safe working areas, especially for damaged structural members such as floor slabs and beams with

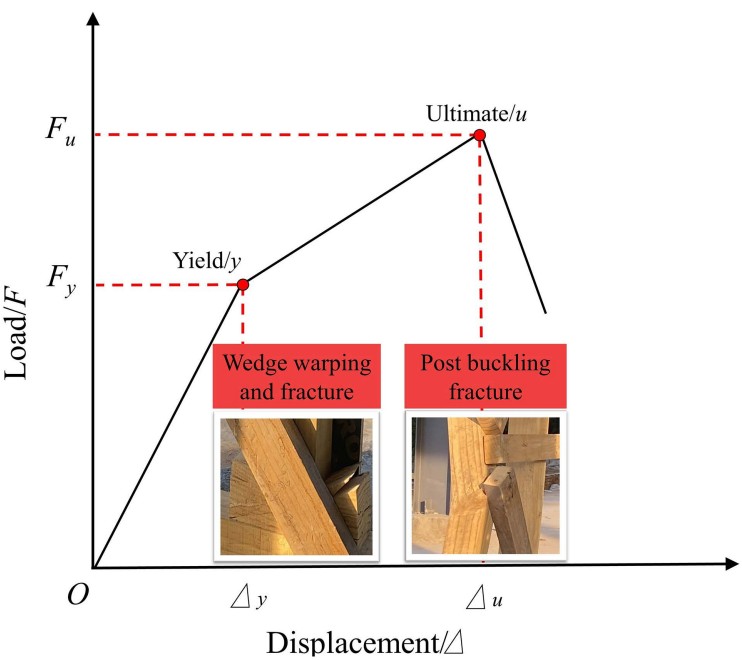

**Fig 12. Compressive bearing capacity model and key early-warning indicators for Type II vertical rescue timber shores.**

localized failure. Their load-transfer mechanism follows a "double funnel" principle: the head plate concentrates upper structural loads, the posts transmit axial forces, and the sole plate distributes these forces to the foundation.

Experimental and numerical results demonstrate that the Two-Post configuration possesses the highest compressive bearing capacity, outperforming the DT-B and DT-A designs, respectively. Load-displacement response analysis indicates a pronounced strength degradation post peak load, underscoring the essential role of lateral bracing elements (e.g., K-braces and gussets) in controlling post deflection and maintaining global stability. The established performance hierarchy, observed failure mechanisms, and proposed safety factor framework are expected to possess broad applicability, though material-specific calibration using local timber species is advised. Selection among shore types entails a strategic compromise: the Two-Post system provides higher load resistance and stability at the expense of complex "K"-bracing fabrication and greater material consumption, making it suited for critical scenarios. In contrast, the DT-A type enables faster deployment due to its simplified construction. The DT-B variant represents an intermediate solution, incorporating a large central gusset that enhances performance relative to DT-A at the cost of additional material and fastening labor.

A primary failure risk for Type II shores arises from load eccentricity. Although a full parametric investigation falls beyond the present scope, experimental and simulation evidence consistently shows that even slight eccentricities severely diminish load capacity. Eccentric loading induces immediate bearing strength reduction, even in the absence of fracture. Should the applied load fall outside the inter-post effective bearing zone, structural capacity approaches zero. Therefore, precise alignment of the resultant force with the shore's centroid is imperative during installation. For the DT-B shore (Fig 13), the optimal loading path is confined to the area bounded by the outer edges of the two posts. While ideal alignment may be difficult to achieve under field conditions, load application should be positioned as close as practicable to this region to ensure effective support.

Plywood gusset failure—notably the bottom half-gussets in DT-type shores and the head/sole plate gussets in the Two-Post system—acts as a critical initiator of systemic instability. Fracture of these components leads to loss of lateral restraint at essential junctions, increasing post eccentricity and lateral deformation, which accelerates progression toward

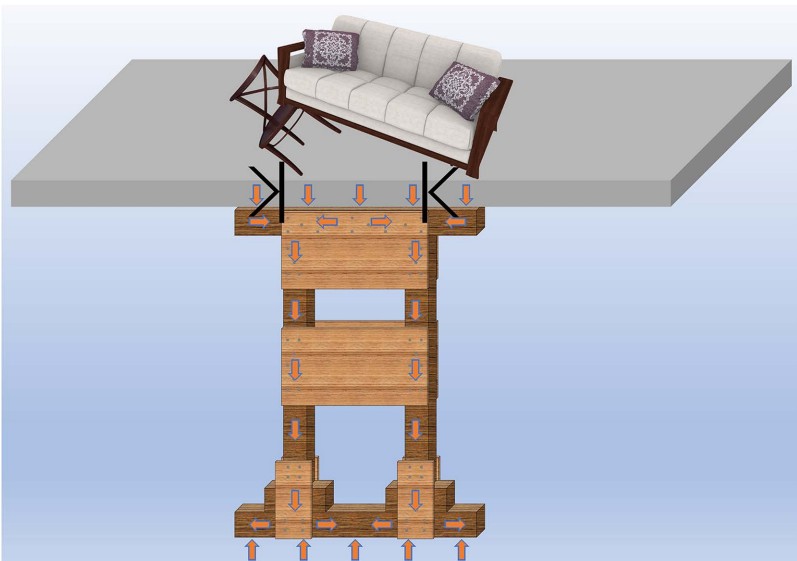

**Fig 13. Working principle and load-bearing schematic of Type II vertical rescue timber shores.**

buckling and structural collapse. Hence, gusset failure is not merely localized damage but a determinative event in the global failure sequence. To improve performance, reinforcement of wedge interfaces—through metal plating or geometric refinement—may retard the initiation of cracking and warping. This would elevate the yield load ($F_y$), prolong the elastic response, and enhance the safety margin via earlier warning. It should be noted, however, that the dominant ultimate failure mode is expected to remain post buckling, though possibly at a moderately increased load level. The validated finite element model presented herein offers a reliable platform for future refinement. Parametric studies examining variables such as gusset thickness, brace angle, and joint detailing can be conducted to balance competing objectives of strength, weight, and material economy—constituting a logical and productive avenue for further investigation.

## 5. Conclusions

This study systematically evaluated the structural performance and failure mechanisms of Type II vertical rescue timber shores through experimentation and numerical simulation, yielding the following key conclusions:

(1) Performance Ranking and Failure Mechanism: The Two-Post shore exhibited the highest compressive bearing capacity (239.0 kN), significantly exceeding that of the DT-B (200.8 kN) and DT-A (190.9 kN) types, with the specific quantitative results (e.g., ultimate loads) being dependent on the timber properties used. All three types exhibited a chain failure path: acoustic emission from wedge cracking → plywood gusset expansion → buckling fracture of posts at the mid-span third. Post fracture was the direct cause of failure.

(2) Bearing Capacity Design Method: Based on the farthest point method for yield point determination, combined with the ASD method and site-specific human-machine-environment factors, a safety factor $K = 2.0$ is proposed. The design bearing capacity is calculated as $F_d = F_u / 2.0$.

(3) Critical Application Criterion: Load eccentricity induces an abrupt ("cliff-edge") reduction in bearing capacity, approaching zero if the load deviates outside the post region. Therefore, ensuring the load acts precisely on the shore's centroid is imperative. For DT-B shores, the load application point must be confined within the area defined by the outer edges of the two posts.

(4) Structural Configuration Recommendation: The Two-Post configuration, benefiting from its "K"-braced frame system which effectively shares horizontal forces, offers advantages in both high bearing capacity and post-failure integrity, and is recommended as the preferred choice. DT-type shores require enhanced lateral constraints to compensate for stability limitations.

In summary, this research elucidates the failure patterns of Type II vertical timber shores, establishes a quantitative design methodology, and defines the safety-critical principle of load centering, providing vital theoretical support for the optimized design and safe application of timber shoring systems in earthquake rescue operations.

## Acknowledgments

The authors gratefully acknowledge the General Research Project (YB2410) of Shandong Earthquake Agency for funding this study.

## Author contributions

**Conceptualization:** feng zhang.

**Data curation:** feng zhang.

**Formal analysis:** feng zhang.

**Investigation:** feng zhang.

**Methodology:** feng zhang.

**Software:** Xiangyang Lu.

**Supervision:** Hao zhang.

**Writing – original draft:** feng zhang.

**Writing – review & editing:** feng zhang.

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
