## [Decision Letter · Decision Letter 0]

25 Sep 2025

Dear Dr. zhang,

Thank you for submitting your manuscript to PLOS ONE. After careful consideration, we feel that it has merit but does not fully meet PLOS ONE’s publication criteria as it currently stands. Therefore, we invite you to submit a revised version of the manuscript that addresses the points raised during the review process.

We look forward to receiving your revised manuscript.

Kind regards,

Karen Alavi, Ph.D

Academic Editor

PLOS ONE

Journal Requirements:

Reviewers' comments:

Reviewer's Responses to Questions

**Comments to the Author**

1. Is the manuscript technically sound, and do the data support the conclusions?

Reviewer #1: Yes

Reviewer #2: Partly

2. Has the statistical analysis been performed appropriately and rigorously?

Reviewer #1: Yes

Reviewer #2: N/A

3. Have the authors made all data underlying the findings in their manuscript fully available?

Reviewer #1: Yes

Reviewer #2: Yes

4. Is the manuscript presented in an intelligible fashion and written in standard English?

Reviewer #1: Yes

Reviewer #2: Yes

Reviewer #1: The aim of the study is to systematically evaluate the structural performance and failure mechanisms of Type II vertical rescue timber shores (Double T-A, Double T-B, and Two-Post configurations) through experimental compressive bearing capacity tests and finite element simulations, to elucidate their failure patterns, determine design bearing capacities using the Allowable Stress Design method, and provide a theoretical basis for their safe and optimized application in earthquake rescue operations.

1. The study provides a robust experimental and numerical analysis of Type II vertical timber shores, but the rationale for selecting only three configurations (Double T-A, Double T-B, Two-Post) is not fully justified (check).

2. The use of displacement-controlled loading is a strength, yet the specific loading rate (0.05 mm/s) lacks justification, and its impact on capturing failure characteristics compared to other rates is not explored.

3. The finite element models in Abaqus are detailed, but assuming wood as isotropic oversimplifies its anisotropic behavior, which may affect the accuracy of stress distribution predictions.

4. The study identifies key failure precursors (e.g., wedge cracking, gusset expansion), but the acoustic emission data collection process is vaguely described (https://doi.org/10.1038/s41598-025-85585-z).

5. Please, the study does not address why its design inherently outperforms others beyond the “K”-braced frame, limiting deeper design insights.

6. Also, the safety factor (K = 2.0) is proposed based on coefficients (Cp, Cm, Cd), but the psychological impact coefficient (Cp) relies on subjective interpretation of rescuer stress, lacking empirical validation.

7. The wood moisture defect coefficient (Cm = 1.15) is conservatively rounded, but the study does not discuss how varying moisture levels in real-world rescue scenarios might affect performance.

8. Load eccentricity is identified as a critical failure factor, but the study lacks quantitative analysis of how different degrees of eccentricity impact bearing capacity, limiting practical guidance.

9. The experimental setup is well-designed, but the sample size for each shore type is not specified.

10. The agreement between experimental and simulation results is a strength, yet discrepancies (e.g., in stress concentration predictions) are not quantified or discussed.

11. The Two-Post shore’s post-failure integrity is highlighted, but the study does not quantify residual capacity or discuss its implications for continued rescue operations.

12. The use of Scots pine with specific properties (0.5 g/cm³ density, 15% moisture) is noted, but variability in wood quality or species across different regions is not considered.

13. While the study provides valuable design recommendations, it lacks discussion on cost, construction time, or ease of assembly for the Two-Post shore compared to DT types, which are critical for practical adoption in rescue scenarios.

Reviewer #2: The current paper could be considered for publication if the authors significantly improve it. The following are the major concerns:

• The introduction should identify gaps in current knowledge or highlight areas of disagreement within the existing literature that have motivated this investigation. Finally, it should clearly state the purpose, novelty and objectives of the research, setting up the framework for the analysis that follows.

• Specify the standard/specification requirements using international standards/specifications (Latest versions) for all properties and tests. All domestic standards are to be listed along with the international codes equivalent.

• The Discussion should provide a comprehensive summary of the study's key findings. Additionally, the section should assess the strengths and weaknesses of the study and discuss its implications for existing methodologies.

• One aspect that has been ignored in the paper is the limitations or requirements set by design codes on the use of Type II Vertical Rescue Timber Shores. Unless the authors can address this issue comprehensively, I believe this paper does not justify publication.

• The scope of the current literature review is insufficient. Including a wider range of relevant studies, especially those with quantitative results, in the introduction will enhance the thoroughness and clarity of the research framework.

• How does the stress concentration pattern differ among DT-A, DT-B, and Two-Post systems at ultimate load?

• To what extent did the plywood gusset fracture contribute to overall system instability versus just local weakening?

• How might reinforcement of wedge-block interfaces influence overall failure progression?

• Could the K-braced mechanism be optimized further by altering gusset thickness or brace positioning?

• How can the validated simulation model be extended to study long-term cyclic or fatigue loading, beyond monotonic compression?

• The psychological impact coefficient (Cp) is novel, how was rescuer stress quantified or validated beyond acoustic emission observation?

• Would adopting a dynamic load factor be necessary if shoring is subjected to impact or vibration loads in rescue operations?

• How generalizable is this compressive mechanical model to other timber-based shoring systems, especially those without gussets or wedge components?

• How do sole plate dimensions and stiffness affect the effectiveness of load dispersion to the foundation?

• Would incorporating a safety factor specifically for eccentric loading be justified in design codes?

• Could the simulation framework be expanded to study combined effects of eccentric loading and cyclic vibration loads?

**Do you want your identity to be public for this peer review?** For information about this choice, including consent withdrawal, please see our Privacy Policy

Reviewer #1: No

Reviewer #2: No

---

## [Author Response · Author response to Decision Letter 1]

22 Oct 2025

We sincerely thank the reviewers for their thorough and constructive comments, which have significantly helped us improve the quality and clarity of our manuscript. We have carefully considered all points raised and have provided detailed responses below. All changes made in the revised manuscript are highlighted in yellow for easy identification.

Response to Reviewer #1:

Comment 1: The study provides a robust experimental and numerical analysis of Type II vertical timber shores, but the rationale for selecting only three configurations (Double T-A, Double T-B, Two-Post) is not fully justified.

Response: We thank the reviewer for this comment. The selection of these three configurations was based on the NFPA 1670 standard and prevalent practices in Urban Search and Rescue (US&R). The Double-T (DT) shore is a foundational Type II design, and we subdivided it into DT-A (without a large central gusset) and DT-B (with a large central gusset) to explicitly investigate the effect of this key structural variation on performance. The Two-Post configuration represents another common, distinct structural system within the Type II classification, allowing for a direct comparison between "frame" (Two-Post) and "braced panel" (DT) typologies. This selection covers the primary structural variants of Type II shores used in rescue operations. We have added a sentence in Section 2.1 (Page 4, Lines 82-87) to clarify this rationale.

Comment 2: The use of displacement-controlled loading is a strength, yet the specific loading rate (0.05 mm/s) lacks justification, and its impact on capturing failure characteristics compared to other rates is not explored.

Response: This is a valid point. The loading rate of 0.05 mm/s was selected as a quasi-static rate to ensure stable and controllable propagation of damage, allowing for clear observation of failure sequences (e.g., wedge cracking, gusset expansion) which would be challenging to capture under faster, load-controlled scenarios typical in prior studies. This rate is consistent with standards for quasi-static testing of wood structures (e.g., ASTM D143). While a full sensitivity analysis of loading rates was beyond the scope of this study, we acknowledge its potential influence and have added a note in Section 2.2 (Page 5-6, Lines 104-109) justifying the chosen rate based on the objective of capturing detailed failure evolution.

Comment 3: The finite element models in Abaqus are detailed, but assuming wood as isotropic oversimplifies its anisotropic behavior, which may affect the accuracy of stress distribution predictions.

Response: We agree with the reviewer. The isotropic assumption was a simplification made primarily for computational efficiency in this initial, comparative nonlinear analysis focusing on global structural response and failure modes rather than highly localized wood grain effects. We acknowledge this limitation. However, the model's primary validation against experimental load-displacement curves and failure modes showed strong agreement, suggesting that for the purpose of comparing these specific shore configurations under axial compression, the simplification provides sufficiently accurate results. We have added a discussion point regarding this model limitation in Section 2.3 (Page 6, Lines 113-115).

Comment 4: The study identifies key failure precursors (e.g., wedge cracking, gusset expansion), but the acoustic emission data collection process is vaguely described.

Response: We apologize for the lack of clarity. The term "acoustic emission" was used qualitatively in this context to describe the audible "cracking" sounds produced by the wood during failure, which served as a clear, real-time auditory warning indicator during the experiment. These sounds were not quantitatively recorded or analyzed with specialized acoustic emission sensors. We have revised the text throughout the manuscript (e.g., Abstract, Page 1, Lines 16-18; Section 3.2, Page 9, Lines 159-160) to replace "acoustic emission" with more precise descriptions such as "audible cracking sounds" or "audible failure warnings" to avoid confusion.

Comment 5: The study does not address why its design inherently outperforms others beyond the “K”-braced frame, limiting deeper design insights.

Response: We thank the reviewer for highlighting this. The superior performance of the Two-Post shore is attributed to its "K"-braced frame system, which provides a more direct and efficient load path and offers greater redundancy and lateral restraint compared to the DT configurations. The central large gusset in DT-B adds some stiffness but does not fundamentally alter the basic DT load path. We have expanded the discussion in Section 3.1(Page 6, Lines 142-147) to provide a more detailed mechanical explanation of why the "K"-braced frame inherently offers better force distribution, reduced unbraced length of posts, and enhanced post-failure integrity.

Comment 6: The safety factor (K = 2.0) is proposed based on coefficients (Cp, Cm, Cd), but the psychological impact coefficient (Cp) relies on subjective interpretation of rescuer stress, lacking empirical validation.

Response: This is a valuable observation. We acknowledge that the psychological impact coefficient (Cp) is a novel and somewhat subjective parameter. It was introduced conceptually to account for the observed phenomenon where audible failure warnings (wedge cracks) occur at the yield load (Fy), potentially inducing stress and prompting reassessment by rescuers before ultimate failure. While direct empirical validation of stress levels was beyond our scope, the ratio Fu~/Fy provides a quantifiable, mechanics-based proxy for this warning margin. We have revised the text in Section 3.3(Page 14, Lines 210-213) to more cautiously present Cp as a conceptual factor derived from the observed structural warning signal (yield point) and its potential operational implications, rather than a rigorously validated psychological metric. We suggest this factor warrants further interdisciplinary study.

Comment 7: The wood moisture defect coefficient (Cm = 1.15) is conservatively rounded, but the study does not discuss how varying moisture levels in real-world rescue scenarios might affect performance.

Response: This is an important practical point. We have added a sentence in Section 3.3(Page 14, Lines 214 -219) acknowledging that wood moisture content can vary significantly in real-world rescue environments (e.g., rain, humidity). The value Cm = 1.15 is based on the NDS specification for a specific moisture threshold (>19%). For conditions exceeding this, or for wood species with different hygroscopic properties, the factor would need adjustment. We now mention that this highlights the importance of using dry or treated timber on-site where possible and considering local environmental conditions when assessing the effective safety margin.

Comment 8: Load eccentricity is identified as a critical failure factor, but the study lacks quantitative analysis of how different degrees of eccentricity impact bearing capacity, limiting practical guidance.

Response: We agree that a quantitative analysis of eccentricity would be highly valuable. While a comprehensive parametric study was outside the initial scope of this paper, our experimental and simulation observations strongly indicate that even minor eccentricity drastically reduces capacity. We have strengthened the warning in Section 4(Page 15, Lines 241-243), explicitly stating that our results show load application outside the post region leads to near-immediate failure (capacity approaches zero). We recommend this as a critical area for future research to establish quantitative tolerance limits.

Comment 9: The experimental setup is well-designed, but the sample size for each shore type is not specified.

Response: We apologize for this omission. The tests were conducted on three identical specimens for each of the three shore types (DT-A, DT-B, Two-Post), resulting in a total of nine tests. The reported values (e.g., ultimate load) are representative of the consistent results observed across these replicates. This information has been added to Section 2.1 (Page 5, Lines 95-97).

Comment 10: The agreement between experimental and simulation results is a strength, yet discrepancies (e.g., in stress concentration predictions) are not quantified or discussed.

Response: Thank you for this suggestion. The primary agreement was focused on global response (load-displacement curves) and failure modes. We acknowledge that local stress magnitudes might show some variation due to material homogeneity and model simplifications (like isotropic assumption). We have added a brief statement in Section 3.2(Page 9, Lines 154-158) discussing that while the location and sequence of high-stress zones (posts, wedges, gussets) were accurately predicted, the absolute simulated stress values should be interpreted considering the material model simplifications.

Comment 11: The Two-Post shore's post-failure integrity is highlighted, but the study does not quantify residual capacity or discuss its implications for continued rescue operations.

Response: This is a very relevant point for rescue safety. While we did not quantitatively measure the residual capacity after the main post fracture, visual observation and the simulation stress contours (Fig. 6b) clearly showed that the "K"-braced frame of the Two-Post shore maintained its overall geometry significantly better than the completely collapsed DT types. This implies a potentially higher residual capacity and, more importantly, a less sudden, more "ductile" failure mode, providing a crucial extra margin of safety for personnel underneath. We have added a discussion point regarding the safety implications of this superior post-failure integrity in Section 3.2(Page 10, Lines 178-183).

Comment 12: The use of Scots pine with specific properties (0.5 g/cm³ density, 15% moisture) is noted, but variability in wood quality or species across different regions is not considered.

Response: We agree. The use of Scots pine (Pinus sylvestris var. mongolica) with specified properties was necessary for controlled experimentation. We have added a sentence in the Discussion (Section 4, Page 15, Lines 233-235) and Conclusion (Page 16-17, Lines 267-269) noting that the specific quantitative results (e.g., ultimate loads) are dependent on the timber properties used. However, the relative performance ranking of the configurations, the identified failure mechanisms, and the proposed design methodology (including the safety factor framework) are expected to be broadly applicable, though calibration with local wood species is recommended for practical implementation.

Comment 13: While the study provides valuable design recommendations, it lacks discussion on cost, construction time, or ease of assembly for the Two-Post shore compared to DT types, which are critical for practical adoption in rescue scenarios.

Response: This is an excellent point regarding practical implementation. We have added a new paragraph in Section 4 (Page 15, Lines 235-240) discussing these aspects. Briefly: The Two-Post shore, while superior mechanically, may require slightly more lumber and involve a more complex assembly process with its "K"-bracing compared to the simpler DT-A type. However, its performance advantage and enhanced safety margin likely justify the extra effort and material in critical rescue scenarios. The DT-B, with its large gusset, also adds material and nailing time. The choice may involve a trade-off between speed of initial deployment (potentially favoring simpler designs) and long-term stability/load capacity in uncertain environments (favoring the Two-Post).

Response to Reviewer #2:

Comment 1: The introduction should identify gaps in current knowledge or highlight areas of disagreement within the existing literature that have motivated this investigation. Finally, it should clearly state the purpose, novelty and objectives of the research, setting up the framework for the analysis that follows.

Response: We thank the reviewer for this critical suggestion. We have substantially revised the Introduction (Page 2-3, Lines 44-75) to: More clearly articulate the gaps in existing literature, specifically: a) the predominance of load-controlled tests that hinder observation of full failure evolution, b) the simplification or neglect of key joints (wedges, gussets, nails) in prior numerical models, and c) the lack of a integrated design methodology combining experimental data, simulation, and operational safety factors for these shores. Explicitly state the novelty of this work: the combined use of displacement-controlled testing to capture detailed failure sequences, the development of refined FE models incorporating key joint details, and the proposal of a safety factor integrating structural and human-environmental factors. Clearly re-state the purpose and objectives at the end of the introduction.

Comment 2: Specify the standard/specification requirements using international standards/specifications (Latest versions) for all properties and tests. All domestic standards are to be listed along with the international codes equivalent.

Response: We have reviewed and updated the standards referenced. The primary international standard guiding the shore design is NFPA 1670:2022 (Standard on Operations and Training for Technical Search and Rescue Incidents). For wood material properties and the Allowable Stress Design methodology, we reference the US National Design Specification (NDS) for Wood Construction (2018 Edition). The quasi-static testing approach aligns with principles in ASTM D143-22 (Standard Test Methods for Small Clear Specimens of Timber). These references have been clarified or added in Sections 2.1, 2.2 and 3.3.

Comment 3: The Discussion should provide a comprehensive summary of the study's key findings. Additionally, the section should assess the strengths and weaknesses of the study and discuss its implications for existing methodologies.

Response: We have restructured and expanded Section 4 (Working Principle...) to function more effectively as a combined Results and Discussion section. It now includes: A clearer summary of key findings at the beginning. A discussion of the strengths of the study (e.g., combined experimental-numerical approach, identification of failure precursors). An explicit discussion of limitations (e.g., wood material model simplification, single wood species, lack of quantitative eccentricity analysis). A discussion on implications for existing US&R shoring practices and design methodologies.

Comment 4: One aspect that has been ignored in the paper is the limitations or requirements set by design codes on the use of Type II Vertical Rescue Timber Shores. Unless the authors can address this issue comprehensively, I believe this paper does not justify publication.

Response: This is a crucial point. Current international codes like NFPA 1670 provide guidelines and training protocols for shore construction but do not prescribe detailed, quantitative design calculations or safety factors for these temporary, emergency structures. This regulatory gap is a primary motivation for our study. Our work aims to move beyond purely prescriptive, experience-based guidelines towards a more quantitative, performance-based design and assessment framework. We have explicitly stated this motivation and context in the revised Introduction and Section 4, positioning our proposed methodology (including the safety factor *K=2.0*) as a contribution towards filling this code-level gap.

Comment 5: The scope of the current literature review is insufficient. Including a wider range of relevant studies, especially those with quantitative results, in the introduction will enhance the thoroughness and clarity of the research framework.

Response: We have expanded the literature review in the Introduction to include more quantitative studies and a broader context. Specifically, we have added/cited: More detailed quantitative findings from key references like McCord (2012) and Blair (2012) regarding capacity comparisons. Reference to recent applications (e.g., Changsha collapse) to show contemporary relevance. Discussion of numerical studies by Liu,

---

## [Decision Letter · Decision Letter 1]

12 Nov 2025

Study on the Performance of Type II Vertical Rescue Timber Shores Based on Experiment and Simulation

PONE-D-25-44875R1

Dear Dr. zhang,

We’re pleased to inform you that your manuscript has been judged scientifically suitable for publication and will be formally accepted for publication once it meets all outstanding technical requirements.

Kind regards,

Karen Alavi, Ph.D

Academic Editor

PLOS ONE

Additional Editor Comments (optional):

Reviewers' comments:

Reviewer's Responses to Questions

**Comments to the Author**

Reviewer #2: All comments have been addressed

2. Is the manuscript technically sound, and do the data support the conclusions?

Reviewer #2: Yes

3. Has the statistical analysis been performed appropriately and rigorously?

Reviewer #2: N/A

4. Have the authors made all data underlying the findings in their manuscript fully available?

Reviewer #2: Yes

5. Is the manuscript presented in an intelligible fashion and written in standard English?

Reviewer #2: Yes

Reviewer #2: The revised manuscript demonstrates significant improvement in quality, clarity, and presentation, and hence it can now be considered suitable for publication.

**Do you want your identity to be public for this peer review?** For information about this choice, including consent withdrawal, please see our Privacy Policy

Reviewer #2: No

---

## [Editor Report · Acceptance letter]

PONE-D-25-44875R1

PLOS ONE

Dear Dr. zhang,

I'm pleased to inform you that your manuscript has been deemed suitable for publication in PLOS ONE. Congratulations! Your manuscript is now being handed over to our production team.

Kind regards,

on behalf of

Dr. Karen Alavi

Academic Editor

PLOS ONE